# Effects of fluid shear stress duration on the mechanical properties of HeLa cells using atomic force microscopy

Xinyao Zhao[1], Xiaolong Zhang[2], Fei Lei[1], Weikang Guo[1], Hui Yu[3], Yaoxian Wang[1]*

1 Department of Gynecological Radiotherapy, Harbin Medical University Cancer Hospital, Harbin, China, 2 College of Shipbuilding Engineering, Harbin Engineering University, Harbin, China, 3 Department of Cardiopulmonary Function, Harbin Medical University Cancer Hospital, Harbin, China

* wyxxs012@126.com

## Abstract

Cellular mechanical properties play a critical role in physiological and pathological processes, with fluid shear stress being a key determinant. Despite its importance, the impact of fluid shear stress on the mechanical characteristics of HeLa cells and its role in the mechanism of tumor metastasis remain poorly understood. This study aims to investigate the effects of varying durations of fluid shear stress on the mechanical properties of HeLa cells, thereby elucidating the mechanical interactions between the fluid flow environment and cancer cells during tumor metastasis. We established an in vitro fluid shear stress cell experimental system and analyzed the flow field characteristics within a parallel plate flow chamber using computational fluid dynamics software. Atomic force microscopy was used to measure the mechanical properties of HeLa cells at different time points under a fluid shear stress of 10 dyn/cm², a value representative of physiological conditions. computational fluid dynamics analysis confirmed the stability of laminar flow and the uniformity of shear stress within the parallel plate flow chamber. The experimental results revealed that with increasing fluid shear stress exposure duration, HeLa cells exhibited a fusiform shape, with a reduction in cell height and a significant decrease in cell Young's modulus. By integrating atomic force microscopy with the in vitro fluid shear stress cell experimental system, this study demonstrates the substantial influence of fluid shear stress on the mechanical properties of HeLa cells. This provides novel insights into the behavior of cancer cells within the in vivo flow environment. Our findings enhance the understanding of cellular mechanical property regulation and offer valuable insights for biomedicine engineering research.

**Data availability statement:** The raw data supporting the findings of this study are fully accessible at figshare through the permanent DOI: [https://doi.org/10.6084/m9.figshare.28490555.v1].

**Funding:** This study was supported by the following funding sources: 1.China Medical Hand in Hand Project Committee, Beijing Medical Award Foundation, Grant Numbers YXJL-2020-0417-0043 and YXJL-2020-0510-0044. WYX received these awards. 2.Beijing Science and Technology Innovation Medical Development Foundation, Grant Number KC2022-JX-0123-06. WYX received this award. 3.Harbin Medical University Graduate Research and Innovation Project, Grant Number YJSCX2023-84HYD. ZXY received this award. For more information on the sponsors, please visit their websites at: 1. China Medical Hand in Hand Project Committee, Beijing Medical Award Foundation: https://www.yxjl.org/cms/html/index 2. Beijing Science and Technology Innovation Medical Development Foundation: https://www.kcyx.org.cn 3. Harbin Medical University: http://www.hrbmu.edu.cn Role of Sponsors/Funders: The funders had no role in study design, data collection and analysis, decision to publish, or preparation of the manuscript.

**Competing interests:** The authors have declared that no competing interests exist.

## Introduction

The convergence of medicine and engineering, known as " Medical-Engineering Integration" is a vital interdisciplinary field that propels the advancement of modern medical technology [1]. In the realm of biomechanics, the study of cellular mechanical properties is particularly pivotal, with fluid shear stress (FSS) being at the forefront of life sciences research [2]. As a crucial parameter of cellular stress in flowing fluids, FSS significantly influences cellular mechanical properties and physiological behaviors [3,4]. Cervical cancer, a significant threat to global women's health, has driven extensive research into its metastasis mechanisms, which is essential for developing innovative treatment strategies. However, the impact of FSS on the mechanical properties of cervical cancer HeLa cells and its role in tumor metastasis remain poorly understood, hindering our comprehensive understanding of the underlying mechanisms of tumor cell metastasis [5].

In recent years, an increasing number of studies have focused on the effects of FSS on the behavior and characteristics of cancer cells. Recent study has demonstrated that by establishing an in vitro model to simulate the effects of fluid shear stress on the adhesion of circulating tumor cells (CTCs) to brain endothelial cells, it was revealed that fluid shear stress can selectively enrich CTCs capable of stably adhering to the brain endothelium, which exhibit a higher potential for brain metastasis [6]. This finding underscores the pivotal role of FSS in cancer metastasis, offering valuable insights for further investigation. Moreover, the influence of fluid flow environments on cellular behavior extends beyond cancer cells. By employing open microfluidic chip technology, researchers have explored the effects of fluid flow conditions in the culture medium on the growth and differentiation of myoblasts (C2C12), further validating the significant contribution of fluid flow environments to cellular differentiation [7]. This study provides a novel perspective on the regulatory mechanisms by which mechanical environments govern cell fate, enriching our understanding of the interplay between physical forces and cellular processes.

It is worth noting that the mechanical properties of cancer cells also play a crucial role during the metastatic process. Agrawal, et al. [8] have demonstrated that the mechanical properties of cancer cells play a crucial role in the metastatic process, and these in vitro findings align with discoveries from in vivo studies. Previous study has demonstrated that the Young's modulus of tumor tissues in vivo is significantly lower than that of normal tissues. This enhanced softness may enable cancer cells to traverse tissue barriers more effectively, thereby promoting their invasion and metastatic potential [9]. Cross et al. [10] measured the mechanical properties of cancer cells in the pleural fluid of cancer patients and found that the Young's modulus of metastatic cancer cells was significantly lower than that of normal cells. This clinical correlation suggests that the mechanical properties of cancer cells may play a crucial role in the metastatic process in vivo.

Previous studies have demonstrated significant differences in mechanical properties between tumor cells and normal cells, particularly in terms of cytoskeletal structure and mechanical characteristics. This is because when cells undergo

pathological changes, their shape and cytoskeleton are altered. These alterations directly lead to changes in cellular mechanical properties (such as deformability, motility, and adhesion ability) and ultimately induce changes in cellular functions (such as gene expression, viability, and proliferative capacity) [11]. However, methodological limitations exist in these studies: some directly analyzed tumor tissues or cells extracted from in vivo environments [12] while others applied fluid shear stress to cells cultured in vitro but failed to adequately simulate dynamic flow conditions present in vivo [13]. Additionally, certain studies investigated cellular mechanical properties under static conditions, resulting in measurements that may not fully reflect the mechanical behavior of cells during actual metastatic processes.

The fluid flow microenvironment plays a crucial role in the occurrence, development, evolution, and clinical treatment of tumors [14,15]. Therefore, how to conduct efficient and systematic in vitro cell experiments while simulating the in vivo tumor microenvironment is vital for obtaining reliable experimental results. In this study, we established an in vitro fluid shear stress cell experimental system and used computational fluid dynamics software to analyze the flow field characteristics within a parallel plate flow chamber, confirming the stability of laminar flow and the uniformity of shear stress within the chamber. The application of this experimental system to study the effects of fluid shear stress on the mechanical properties of cancer cells provides new insights into the behavior of cancer cells during metastasis.

Advances in experimental bioengineering have enabled researchers to directly and real-time probe and manipulate individual cells and molecules mechanically. Hao et al. [16] reviewed the measurement methods and applications of single-cell mechanical properties, such as elastic modulus and shear modulus. They explored the use of single-cell mechanical characterization in cell sorting, disease diagnosis, and drug screening, providing valuable insights for cellular mechanics research and the development of biomedical tools. The invention of atomic force microscopy (AFM) has provided a powerful new tool for the study of cellular mechanical properties [17], enabling precise quantitative measurement of cellular mechanical responses and characteristics [18,19]. This study designed an in vitro fluid shear stress cell experimental system, applying a parallel plate flow chamber model to generate a broad range of uniform shear forces on cervical cancer cells, allowing for the collection of a sufficient number of cells for studies on molecular signaling and gene expression [20–22]. By integrating the in vitro fluid shear stress cell experimental system with AFM technology [23], we have achieved the application of fluid shear stress during in vitro cell culture while ensuring that the cellular culture and mechanical environments are consistent with those inside the human body.

The experimental results indicate that the duration of FSS exposure significantly affects the mechanical properties of HeLa cells, including changes in cell morphology and a decrease in Young's modulus. These findings provide new insights into the behavior of tumor cells in the in vivo flow environment and offer new methods and perspectives for studying the interactions between cancer cells and the extracellular microenvironment during tumor metastasis. This discovery not only enhances our understanding of the regulatory mechanisms of cellular mechanical properties but also provides important information for related research in the field of bio-medical engineering, holding broad and positive significance for exploring the intrinsic mechanisms of life activities.

## Materials and methods

### Cell culture and sample preparation

The present experiment utilized the HeLa cell line of cervical cancer, sourced from the Chinese Academy of Sciences' cell repository. Cells were cultured in DMEM-H medium (BL304A, Biosharp, China), which was supplemented with 10% fetal bovine serum (FSP500, ExCell Bio, China) and 1% penicillin-streptomycin solution (P1400, Solarbio, China) in DMEM medium. DMEM-H was chosen as the culture medium because it is the standard for HeLa cell culture. Its high glucose concentration (4.5 g/L) meets the high metabolic demands of HeLa cells, better supports their growth and proliferation, prevents stress responses caused by nutrient deficiency, and maintains energy metabolism balance under shear stress conditions.

All cells were plated in culture dishes (Bkmamlab, China) and cultivated in an incubator (Shanghai Yiheng Scientific Instrument Co., Ltd., model BPN-50CH(UV)) under conditions of 37°C, 5% $CO_2$, and 95% air.

To facilitate cell adhesion, proliferation, and growth, coverslips treated with poly-L-lysine (Beyotime Biotechnology, China) were employed in the experiment. Poly-L-lysine, a synthetic molecule, significantly enhances the adhesiveness of cells to the culture surface. The experimental procedure is as follows:

1. Immerse clean coverslips in a diluted solution of poly-L-lysine and incubate at room temperature for 30 minutes.

2. After incubation, place the coverslips in a biosafety cabinet and sterilize them with ultraviolet light for 24 hours.

3. Following sterilization, place the coverslips into culture dishes.

4. During cell passaging, add the cell suspension to the coverslips and let them stand for 5 minutes to promote initial cell adhesion.

5. Add appropriate culture medium to the culture dishes.

6. Place the culture dishes in the incubator to maintain optimal culture conditions.

7. When the cells have grown to approximately 70%-100% confluence on the coverslip surface, remove the coverslips in preparation for in vitro fluid shear stress cell experiments.

## Parallel-plate flow chamber systems

To uniformly and extensively apply laminar shear stress to cells, this study designed a parallel-plate flow chamber with dimensions of 70 mm in length, 24 mm in width, and 12 mm in height [24]. The chamber features four threaded holes on the top, which are connected to a transition joint via threads. The transition joint is designed with a conical taper at the top to facilitate connection with the connecting tube. Inside the transition joint, there is a through-hole with a diameter of 1.6 mm. The left and right transition joints are designated for liquid inflow and outflow, respectively, while the upper and lower ones are connected to vacuum tubes. A sealing gasket, measuring 70 mm in length and 24 mm in width with an inner square hole of 41 mm by 10 mm, is placed between the chamber's bottom and a glass slide, with a thickness of 3 mm. During experimental operations, the liquid first enters the transition expansion slot from the left connecting tube, then forms a parallel flow chamber through the rectangular aperture on the sealing gasket, and finally flows into the transition expansion slot on the opposite side before exiting the chamber. The three-dimensional model of the entire flow field was constructed using SolidWorks 3D modeling software, based on the parameters earlier described. For the purpose of flow field dynamics analysis, the internal flow region of the parallel-plate flow chamber was extracted and exported as a STEP file. To focus on the dynamic analysis of shear stress at the bottom wall of the flow chamber and to enhance computational accuracy, the model was imported into Hypermesh software for mesh generation. Twenty layers of mesh were arranged in the thickness direction of the flow region to more precisely capture variations in wall shear stress (refer to Fig 1a and 1b). After mesh generation, the resulting mesh file was imported into ANSYS Workbench 2019 for further analysis and simulation [25].

## Fluid shear stress calculation

Based on the fluid shear stress intensity calculation formula and the dimensions of the designed parallel plate flow chamber, the relationship between shear stress and flow rate can be determined, with the flow rate input serving as the condition for simulation input.

We employed the classical fluid shear stress theoretical formula to calculate the wall shear stress in the parallel plate flow chamber [26]. This formula assumes that the fluid is Newtonian and the flow is steady. In the actual flow chamber,

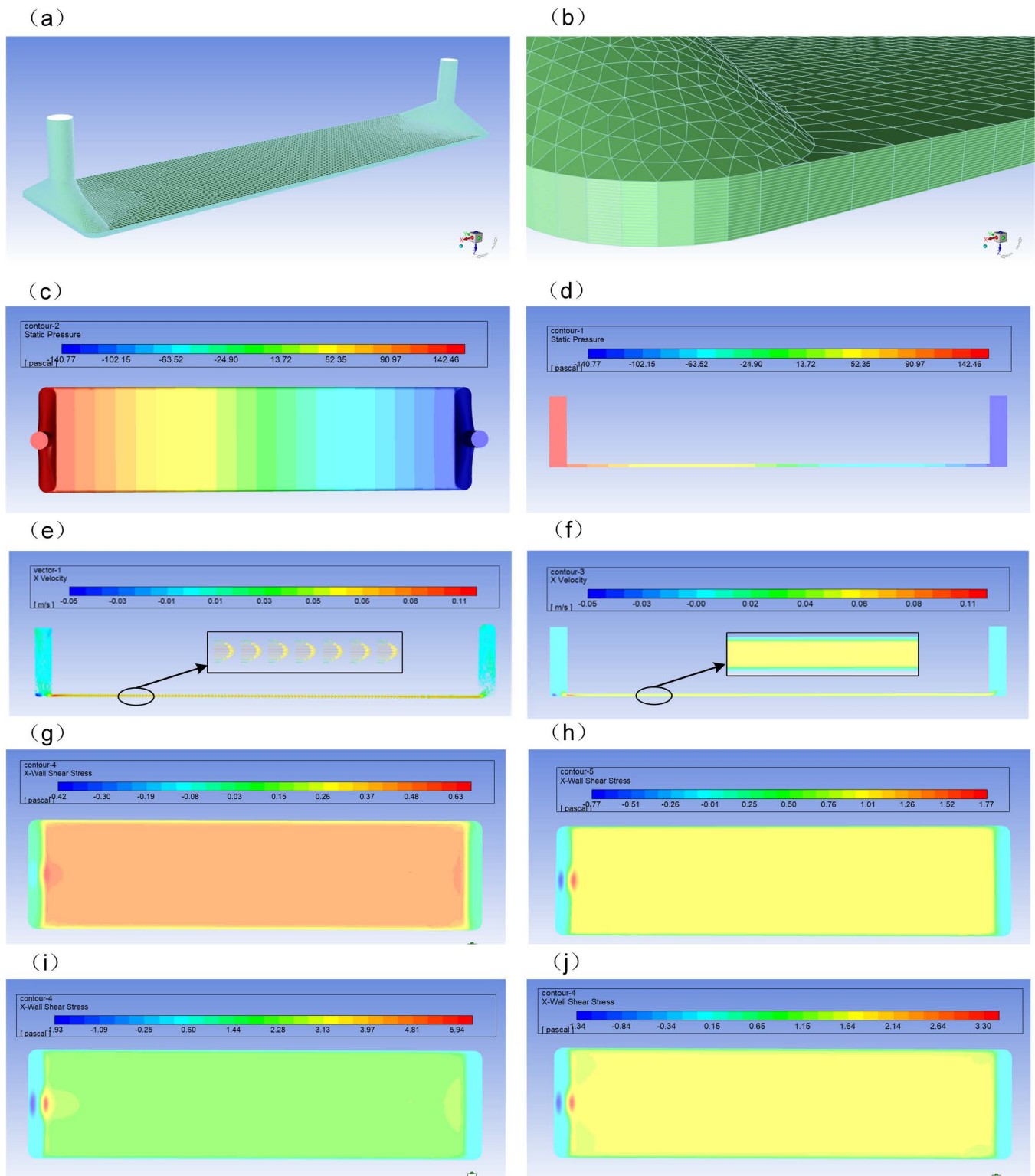

**Fig 1. Computational Fluid Dynamics results for a parallel plate flow chamber based on FLUENT.** (a) Global finite element mesh division of flow field. (b) Local finite element mesh. (c) Global flow field pressure distribution nephogram. (d) Z = 0 characteristic surface pressure distribution nephogram.(e) Flow field velocity vector distribution nephogram.(f) Flow field velocity distribution nephogram.(g-j) Fluid shear stress nephogram for four working conditions (5、10、15 and 20dyn/cm²).

due to the finite width and height of the channel, there is a certain velocity gradient variation at the boundaries of the channel. However, when b/a > 20, these boundary effects become negligible, and more than 85% of the surface is exposed to a uniform wall shear stress.

1) Fluid Shear Stress Calculation Formula [27]:

$$\tau_w = 6\mu Q/a^2 b$$

$\tau_w$: Shear stress, in dynes per square centimeter (dyn/cm²);
$\mu$: Fluid viscosity, in poise (P);
$a$: Channel height, in centimeters (cm);
$b$: Channel width, in centimeters (cm);
$Q$: Liquid flow rate, in milliliters per second (ml/s).
    For this example, an input shear stress of 10 dyn/cm² used, and according to the calculation formula, the required flow rate is found to be 6.45 ml/min.

2) Boundary Condition Settings

The inlet of the flow field is set as a velocity inlet with a velocity of 0.075 m/s. The outlet of the flow field is set as an outflow boundary. The remaining boundaries of the flow field are set as walls with no-slip boundary conditions between the solid walls and the fluid.

3) Computational Model Settings

After setting the aforementioned conditions, the Reynolds number at the inlet can be calculated. The Reynolds number between the two parallel plates confirms that the entire computational domain is in a laminar flow state, with no turbulence present. Therefore, the computational model is set to Laminar, indicating a laminar flow model.

**In vitro fluid shear stress cell experimental system**

In this study, we have developed a cell experimental system designed to simulate the in vivo cellular environment [28]. The system employs a BPN-50CH(UV) model carbon dioxide incubator from Shanghai Yiheng Scientific Instrument Co., Ltd., which is capable of maintaining a constant temperature at 37°C, a stable level of carbon dioxide at 5%, and a high relative humidity of 95%, thus providing optimal conditions for in vitro cell culture. To apply fluid shear stress, we have modified the incubator by drilling holes for the passage of connecting tubes and sealing them to ensure the internal environment remains stable. The liquid used in the parallel-plate flow chamber is DMEM-H medium, supplemented with 10% fetal bovine serum and 1% penicillin-streptomycin solution. Furthermore, these connecting tubes allow for the circulation of fluid both within and outside the incubator, ensuring that cells are cultured in a consistently suitable environment.

**Method for measuring cellular morphology and mechanical properties using atomic force microscopy (AFM)**

This study employed an atomic force microscope (AFM), the Dimension Icon model from Bruker Corporation, U.S.A., equipped with MLCT-C type silicon nitride cantilevers featuring a tip curvature radius of 20 nm, to assess cellular morphology and mechanical properties following exposure to fluid shear stress. The AFM probe, through contact with the sample, leverages the deflection of the micro-cantilever to measure force magnitude, with an integrated optical measurement system and piezoelectric controller to facilitate accurate scanning and indentation experiments [29].
    Morphological imaging of cells was conducted in contact mode, capturing at a resolution of 256 lines per frame, with a scanning rate of 0.3 Hz over an 80 μm range, at a tip velocity of 40 μm/s. The setpoint was maintained at 0.3nN, and the scanning rate for imaging was 0.2 Hz, with the acquisition of a single image taking approximately 20 minutes. Young's

 

modulus measurements of the cells were derived from force-displacement curves obtained in force volume mode. Each cellular region was measured in triplicate, with each set consisting of 256 curves over an area of 5 x 5 µm², scanned at a rate of 0.5 Hz. The loading-unloading cycle was performed at a speed of 5 µm/s, completed within 2 seconds, with each force curve comprising 1024 data points. A maximum load of 3 nN was applied to ensure the cells remained within their elastic limit.

## Data analysis

Analysis of force curves obtained from Atomic Force Microscopy (AFM) indentation experiments facilitates the extraction of cellular Young's modulus. The raw force curves are comprised of approach and retraction curves. Typically, the Young's modulus is derived from the analysis of the approach curve, with the retraction curve primarily used to assess cellular adhesion properties. By converting the approach curve into an indentation curve based on the contact point and fitting the indentation curve with the Sneddon model, we can determine the cellular Young's modulus.

Currently, the main models available for extracting the Young's modulus of cells are the Hertz-Sneddon, JKR (Johnson-Kendall-Roberts), and DMT (Derjaguin-Muller-Toporov) models [30]. The Hertz-Sneddon model neglects the forces in the contact area between the tip and the cell, such as electrostatic forces, adhesive forces, and frictional forces. In contrast, the JKR and DMT models consider the adhesive forces within the contact area. The JKR model is suitable for situations where the tip is large and the adhesive force between the tip and the sample is significant, while the DMT model is applicable when the tip is small and the adhesive force is minimal. However, the most widely used model in practice is the Hertz-Sneddon model [31]. The Hertz-Sneddon model is based on a series of assumptions about the sample being probed, such as isotropy, homogeneity, linear elasticity, axisymmetry, infinite thickness, and smooth surface [32]. Although cells do not strictly meet these conditions, studies have shown that the Hertz model is applicable when the indentation depth is less than 10% of the sample thickness [33].

Therefore, this study employs the Sneddon model to fit the indentation curves, thereby obtaining the cellular Young's modulus [30].

The Sneddon formula is:

$$F_{Sneddon} = \frac{2E\delta^2 \tan\theta}{3(1-v^2)}$$

In this equation,
$F$ represents the applied load by the probe,
$E$ is the Young's modulus of the sample,
$\theta$ is the semi-aperture angle of a conical tip,
$\delta$ is the indentation depth,
$v$ is the Poisson's ratio of the sample (which is commonly considered to be approximately 0.5 for living cells).

As depicted in Fig 2, the AFM probe is controlled to perform an indentation assay on a HeLa cell to record force curves for measuring cellular Young's modulus.

Statistical analysis. GraphPad Prism9.0 software was used for statistical analysis of the data. One-way ANOVA of variance followed by Tukey's test was used to compare the data between different groups. Differences were considered significant at $P<0.05$.

## Results

### Parallel plate flow chamber fluid dynamics calculation results and analysis

This study employed Fluent software to conduct numerical simulations of the flow field to obtain the pressure distribution within the flow field [34–36]. By extracting the mid-plane section along the length direction and the bottom surface section

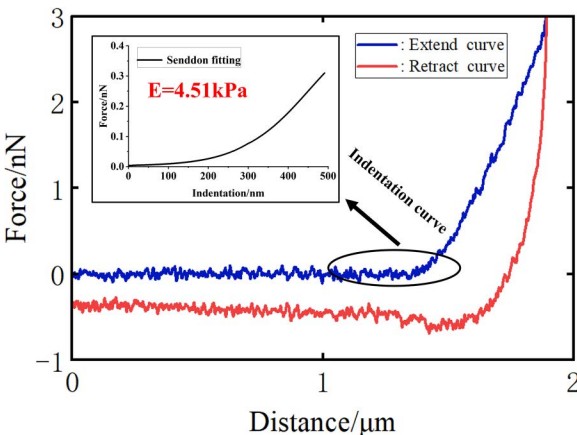

**Fig 2. AFM probe is controlled to perform indentation assay on a Hela cell to record force curves for measuring cellular Young 's modulus.**

of the flow region as characteristic planes, we analyzed the distribution characteristics and trends of flow field pressure, velocity, and wall shear stress.

The computational results reveal the pressure distribution within the flow field, as depicted in Fig 1 (Fig 1c for the overall pressure distribution and Fig 1d for the Z = 0 characteristic plane pressure distribution): The overall pressure distribution within the flow region indicates that pressure in the Y direction remains relatively constant, with the pressure gradient primarily observed in the X direction. The Z = 0 characteristic plane pressure distribution illustrates that pressure decreases gradually along the X direction, with the zero pressure plane (reference pressure) at X = 0, resulting in a negative pressure at the outlet. The pressure difference drives the fluid through the flow chamber, with the maximum pressure value reaching approximately 142.46 Pa. Therefore, subsequent analyses will focus on the components in the X direction (the flow direction of the fluid in the flow field).

Furthermore, the velocity distribution within the flow field was analyzed using velocity vector diagrams and velocity distribution clouds (as shown in Fig 1e and 1f): The velocity vector distribution cloud and velocity distribution cloud on the Z = 0 characteristic plane show that the flow in the central region is quite stable, with relatively high and uniformly distributed velocity. The velocity gradient is significant at the inlet but quickly becomes steady. The wall shear stress is directly proportional to the velocity gradient, with the central region velocity being greater than the boundary velocity, resulting in a stable and smooth flow that generates a velocity gradient, thereby producing fluid shear stress on the boundary walls. The X-direction velocity distribution cloud indicates that the flow region is in standard laminar flow, with a stable flow pattern throughout the process.

The computational results obtained for the bottom surface (Y = 0) characteristic plane of the flow direction show the fluid shear stress distribution along the X direction, as depicted in Fig 1j's fluid shear stress distribution cloud diagram. The fluid shear stress along the X direction ranges from 9.95 to 10.01 dyn/cm², which is consistent with the theoretical calculation results, confirming the authenticity and reliability of the numerical simulation results in this study.

To explore the variation of wall shear stress in the flow field under different flow velocities, fluid shear stresses of 5, 10, 15, and 20 dyn/cm² were analyzed. According to the theoretical calculation formula for fluid shear stress, the corresponding flow Rate are 0.075 ml/s, 0.15 ml/s, 0.225 ml/s, and 0.3ml/s, respectively. The distribution clouds of wall fluid shear stress at different flow velocities (Fig 1g-j) indicate that the distribution trend is essentially the same, with the magnitude being directly proportional to the flow velocity and consistent between theoretical calculations and numerical results. This result validates that the parallel plate flow chamber designed in this study is suitable for practical experimental applications.

In this study, the flow rate Q corresponding to four different shear stress levels was determined based on theoretical calculation formulas. The shear stress values were 5, 10, 15, and 20 dyn/cm², with corresponding flow rates of 0.075, 0.15, 0.225, and 0.3 ml/s, respectively. For each flow rate parameter, both the theoretical shear stress values and the numerical simulation results were calculated. The specific results are presented in the Table 1 below:

Theoretical and numerical fluid shear stress values were calculated for different flow rates (Q). The error between theoretical and numerical results was less than 1% for all tested flow rates.

As can be seen from Table 1, the error between the theoretical and numerical results is less than 1% in all cases. This demonstrates that our method for calculating fluid shear stress exhibits high accuracy and can be reliably utilized in fluid shear stress experiments.

## Applications of in vitro fluid shear stress experimental systems

In this study, we established an in vitro fluid shear stress cell experimental system (Fig 3a) and designed a parallel-plate flow chamber (Fig 3b) to simulate the effects of fluid shear stress on cells, based on the results of flow field dynamics analysis. The experimental system comprises a $CO_2$ incubator, a parallel-plate flow chamber, a vacuum pump (Taizhou Fujian Tool Co., Ltd., model XVP750, capable of reaching a vacuum level of -98 kPa), a peristaltic pump (Baoding Rongbai Constant Flow Pump Manufacturing Co., Ltd., model BT301LY), culture dishes, and connecting tubing (Nanjing Runze Fluid Equipment Co., Ltd., model 14# peristaltic pump silicone tubing, with an inner diameter of 1.6 mm and an outer diameter of 4.8 mm). The system connects the culture dishes and the parallel-plate flow chamber inside the $CO_2$ incubator to the external vacuum pump and peristaltic pump via connecting tubing, forming a closed-loop circuit for precise control of fluid shear stress and maintaining a stable cell culture environment (Fig 3c). The parallel-plate flow chamber used in the experiments is depicted in Fig 3d.

Before the experiment, all components of the parallel-plate flow chamber and the associated experimental equipment were sterilized at high temperatures and then stored at room temperature for later use. HeLa cells, derived from cervical cancer, were transferred onto glass slides and cultured in petri dishes until they covered more than 70% of the slide's surface area. After observing the cells under a microscope and confirming their good condition, we commenced the experimental preparations. During the experiment, the inflow and outflow tubes and the vacuum line of the parallel-plate flow chamber were appropriately connected, and the glass slide was assembled with the flow chamber and a silicone gasket. The vacuum pump was then activated to evacuate the air from the internal vacuum slot of the chamber, creating a sealed environment under atmospheric pressure, which ensured the sealing integrity and uniformity of the flow channel. Following this, the peristaltic pump, which had been pre-set to the correct speed and direction, was turned on. This pump drove the culture medium to flow through the parallel-plate flow chamber via the connecting tubing, subjecting the cells to a uniform and unidirectional fluid shear stress.

After the application of fluid shear stress in the cellular experiment system, Atomic Force Microscopy (AFM) was employed to acquire cell morphology and to measure the mechanical properties of the cells. The AFM model used was the Dimension Icon from Bruker Inc., USA, as shown in Fig 3e. The AFM probe model used was the MLCT-C from Bruker

Table 1. Comparison of theoretical and numerical results for fluid shear stress.

| Flow Rate Q(ml/s) | Theoretical Fluid Shear Stress $\tau_w$(dyn/cm²) | Numerical Fluid Shear Stress $\tau_w$(dyn/cm²) | Error |
|---|---|---|---|
| 0.075 | 5 | 5.01 | <1% |
| 0.15 | 10 | 10.04 | <1% |
| 0.225 | 15 | 15.05 | <1% |
| 0.3 | 20 | 20.05 | <1% |

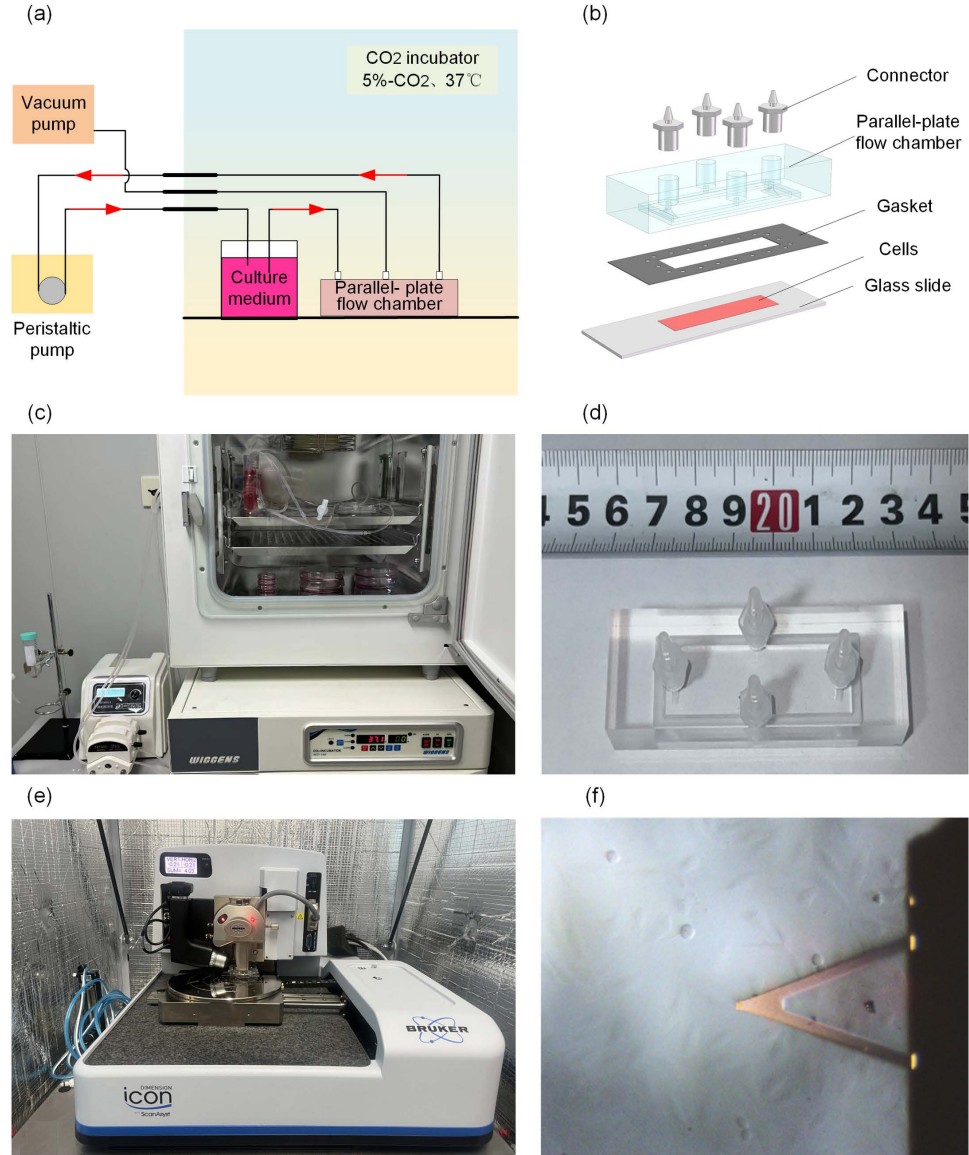

**Fig 3. The experimental technology combining a fluid shear stress experimental system with Atomic Force Microscopy (AFM).** (a) Schematic diagram of a fluid shear stress experimental system. (b) Assembly diagram of a parallel-plate flow chamber. (c) Experimental Apparatus of Fluid Shear Stress System. (d) Parallel Plate Flow Chamber. (e) Atomic Force Microscopy (AFM). (f) AFM probe is controlled to measure single living Hela cells under the guidance of optical microscope.

Inc., USA. Fig 3f illustrates the probe positioned above the cell in preparation for morphological and indentation measurement experiments.

## The effect of fluid shear stress on the mechanical properties of HeLa cells

This study aims to explore the impact of the duration of fluid shear stress on the mechanical properties of cervical cancer HeLa cells. We utilized a self-designed fluid shear stress cell experimental system to simulate the in vivo blood flow environment. To eliminate the impact of cell passage batch variations on the measurement results, all experiments were

conducted using cells from the same passage. The cells were cultured under identical conditions, and the application of fluid shear stress in the in vitro system was performed in the same environment. Additionally, five independent experiments were carried out to ensure the reliability of the results. A parallel plate flow chamber was installed in a $CO_2$ incubator, and the speed of the peristaltic pump in the fluid shear stress cell experimental system was adjusted to apply a fluid shear stress of 10 dyn/cm² to HeLa cells. The cells were exposed to fluid shear stress for 5, 15, 30, and 60 minutes and then compared with a resting control group.

When employing AFM to characterize the mechanical properties of cells, the procedure was initiated by guiding the AFM probe under an optical microscope to acquire the topography of a group of cells located at the center of the culture dish, followed by the collection of force curves from these cells. Subsequently, the AFM probe was precisely controlled to perform individual measurements and obtain force curves from four additional groups of cells in the vicinity of the initially measured cells.

Optical microscopy was first used to capture images of cell morphology under different durations of fluid shear stress. As shown in Fig 4a, the analysis of these images revealed that as the duration of fluid shear stress increased, the cell morphology tended to become more fusiform and stable.

Subsequently, atomic force microscopy (AFM) was employed to scan cell morphology images [37], observing the morphological state of HeLa cells under various durations of fluid shear stress. The AFM's integrated optical microscope was used to position the probe above the HeLa cells, and cell scanning morphology acquisition was conducted in contact mode. The high-resolution images obtained from AFM scanning provided a complete morphology of individual cells, allowing us to collect mechanical property parameters from various regions of the cells. A cell contour graph across the highest point of the cell, i.e., passing through the cell base and the nucleus, was extracted from the measured cell morphology to analyze the effect of fluid shear stress duration on cell height (Fig 4b) [38].

The data showed that under static conditions, the cell morphology height and contour graphs indicated that the cells were plump, with a maximum height of 3.59 micrometers. As the duration of fluid shear stress increased, the height of HeLa cells gradually decreased to 3.59 micrometers, 3.32 micrometers, 3.01 micrometers, 2.42 micrometers, and 1.96 micrometers, respectively (Fig 4c).

The experimental results indicated that as the duration of fluid shear stress increased, the cell morphology gradually changed from round to fusiform, and the flatness of the cells increased. Quantitative analysis of cell height parameters revealed that cell height significantly decreased with increasing duration of shear stress, suggesting that the stability of cell morphology is closely related to the duration of fluid shear stress. These findings provide a new perspective for understanding the role of fluid dynamics in cell morphological changes and offer important experimental methods and data for the study of cell-fluid interactions in biomedical engineering.

Utilizing atomic force microscopy (AFM), cell morphological and noise maps were obtained (Fig 5a and 5b), which allowed us to pinpoint the highest point of the cells, specifically the nuclear region. At this location, a 5x5 µm^2 indentation experiment was conducted. Employing the FORCE VOLUM mode of the AFM, we performed indentation experiments on the nuclear area of the cells. The force-displacement curves derived from these experiments were analyzed using the Sneddon model to fit the data, yielding a map of the cells' Young's modulus (Fig 5c).

We collected force curves from five groups of cells under each experimental condition and analyzed the results from multiple sets of force curves. For each group of cells, 20 force curves were extracted, and the results were integrated as follows:

The data extracted from the cellular Young's modulus maps were compared, as shown in Fig 6. The experimental results indicate that under identical fluid shear stress conditions, the Young's modulus values of the cells tend to decrease with increased duration of exposure. As the duration of fluid shear stress application is extended, there is a significant reduction in the cells' Young's modulus, demonstrating that fluid shear stress significantly affects the mechanical properties of cells. This effect may be associated with the reorganization of the cytoskeleton, alterations in the fluidity of the cell membrane, and changes in intracellular signal transduction.

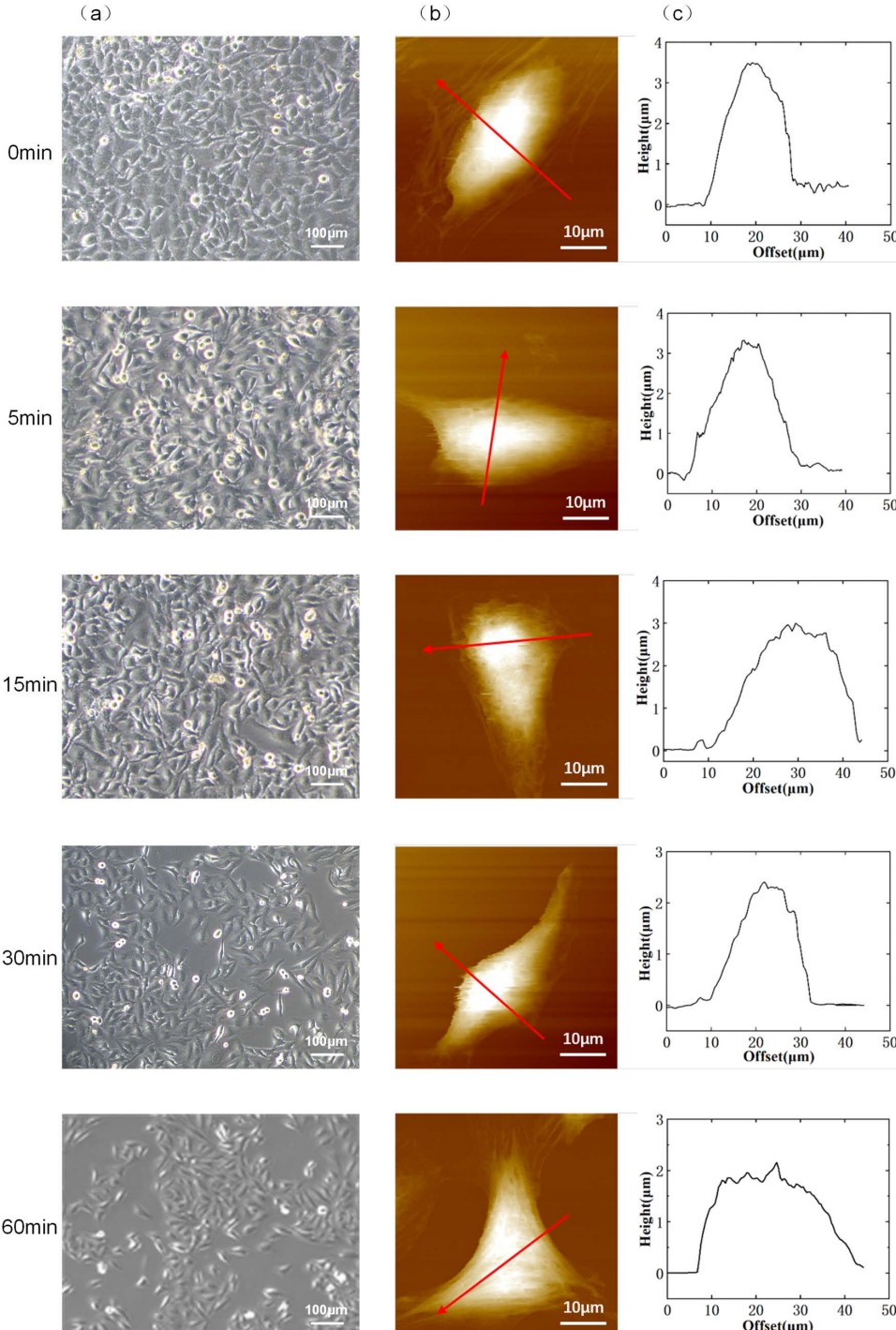

**Fig 4. Comparative Analysis of Hela Cells Cultured under Static Conditions and Exposed to Fluid Shear Stress.** (a) Optical microscopy images comparing the effect of fluid shear stress duration on cell morphology. (b) Atomic Force Microscopy (AFM) height maps of cells under different fluid shear stress durations. (c) Cell height curve under different fluid shear stress over time.

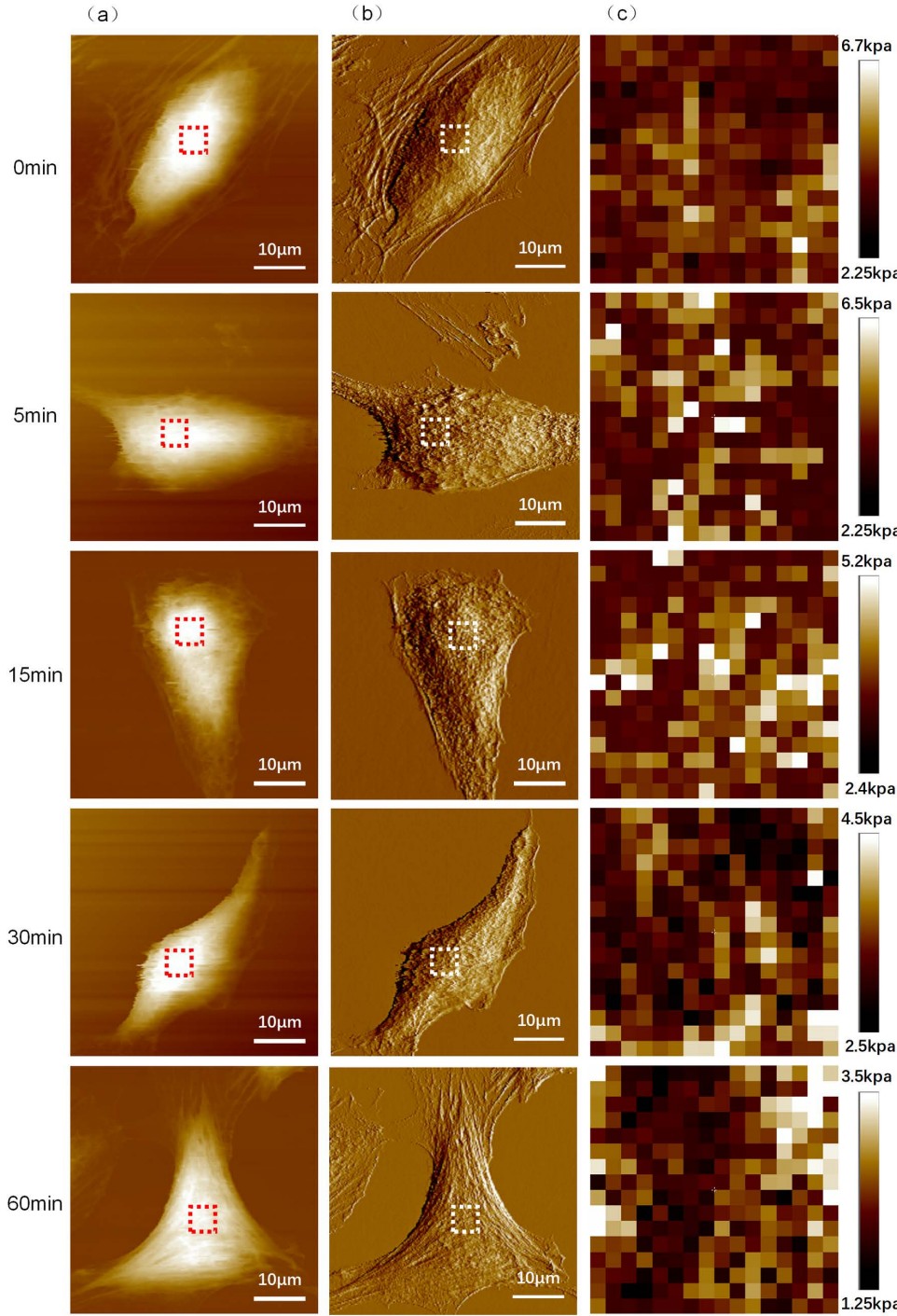

**Fig 5. Measuring results of Hela cells subjected to different durations of fluid shear stress by Atomic Force Microscopy (AFM)** (a) AFM topography of Hela cell. (b) Corresponding deflection images of the margin of Hela cell. (c) Young's Modulus Graph obtained by Atomic Force Microscopy (AFM).

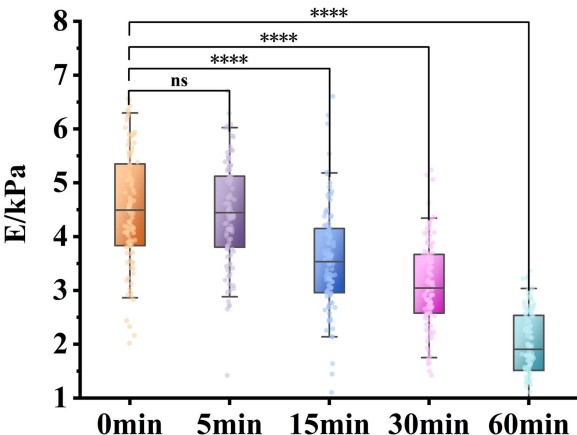

**Fig 6. Measuring the Young's modulus of Hela cells subjected to different durations of fluid shear stress by Atomic Force Microscopy (AFM).**

The findings of this study offer new insights into how fluid shear stress regulates the mechanical properties of cells, which is of significant importance for research in cell biology, biomedicine engineering, and the understanding of related pathological mechanisms. Moreover, these discoveries could also impact the development of new biomaterials and cell therapy strategies, particularly in fields where precise control of cellular mechanical properties is required.

## Discussion

This study utilized a custom-designed in vitro fluid shear stress (FSS) cell experimental system to investigate the impact of fluid shear stress on HeLa cells, providing novel insights into the biomechanical dynamics of cervical cancer cells. Our findings highlight the significant influence of FSS on cellular mechanical properties. Computational fluid dynamics (CFD) analysis of the refined parallel-plate flow chamber ensured a stable and uniform flow field, critical for precise biomechanical analysis.

Studies have shown that cancer cells are exposed to complex mechanical microenvironments, which influence their morphology, structure, and mechanical properties. During the process of cancer cells migrating from the primary site to metastatic sites, lower stiffness facilitates their motility and metastatic potential [39]. Based on the methodology established in this study, FSS was set to 10 dyn/cm² in an in vitro fluid shear stress experimental system, and applied for 0, 5, 15, 30, and 60 minutes, respectively. Cell force curve measurements were conducted on five groups of HeLa cells under each condition. Using the Sneddon model to fit the results, the Young's modulus of HeLa cells was calculated. The analysis revealed that as the duration of fluid shear stress increased, the Young's modulus of HeLa cells showed a significant decreasing trend, with a more pronounced reduction observed at longer exposure times. This result indicates that fluid shear stress has a notable impact on the mechanical properties of HeLa cells. Research has shown that changes in the Young's modulus of cells are closely related to the increased metastatic potential of cancer cells. Lower cell stiffness is generally associated with more aggressive cancer cell behavior, likely due to the reorganization of the cytoskeleton under fluid shear stress [40]. Cancer cells with higher metastatic potential tend to have a lower Young's modulus. Reduced cell stiffness may facilitate the ability of cancer cells to penetrate blood vessel walls or the extracellular matrix (ECM), thereby promoting distant metastasis [41].

FSS-induced cytoskeletal reorganization is a key mechanism by which cells adapt to mechanical forces. The reorganization of the cytoskeleton leads to changes in the mechanical properties of cells. Studies have found that breast cancer cells, after experiencing shear stress in the tumor microenvironment (TME), exhibit increased cell area, reduced roundness, and a more motile phenotype. These cells are more proliferative and demonstrate higher resistance to paclitaxel

[42], indicating that fluid shear stress promotes cancer cell proliferation, invasive potential, and chemoresistance. Recent research utilizing polydimethylsiloxane (PDMS)-based templates to create micropores observed that cells under mechanical stimulation exhibited reduced height and flattened morphology, accompanied by a decrease in Young's modulus [43]. These changes further influence cell functions, including migration and invasion capabilities. Additionally, Quesada et al. [44] found that under fluid shear stress of 10 dyn/cm², the expression of the proliferation marker Ki67 in the MCF-7 cell line was enhanced, suggesting that FSS may promote a more aggressive phenotype through increased protein phosphorylation. These findings demonstrate that FSS can alter the mechanical properties of cells, thereby affecting cell migration, invasion, and intracellular signaling pathways.

Although this study provides significant insights into the mechanical properties of cancer cells, certain limitations remain. In vitro experiments are typically conducted under static or single-variable conditions, making it difficult to fully replicate the complex TME in vivo, as well as the dynamically changing stiffness and composition of the ECM, both of which interact with cancer cells and influence tumor behavior [45]. Therefore, future research needs to develop experimental models that more closely resemble the in vivo environment to better simulate the dynamic characteristics of the tumor microenvironment. Integrating in vitro findings with clinical data will enable a more comprehensive understanding of the role of cancer cell mechanical properties in cancer metastasis and provide a foundation for developing new therapeutic strategies.

The detection of cellular mechanical properties under fluid shear stress environments based on AFM holds broad and positive significance for the life sciences. The microenvironment of bodily fluid flow plays a crucial role in the initiation, progression, evolution, and clinical treatment of tumors [46]. Therefore, the in vitro fluid shear stress experimental system and the methodology for probing cellular mechanical properties developed in this study not only contribute to understanding the impact of fluid shear stress on the mechanical properties of cancer cells but also have wide-ranging applications in addressing related scientific questions in the life sciences.

## Conclusion

In summary, this study demonstrates that FSS significantly alters the mechanical properties of HeLa cells. The integration of AFM with the in vitro fluid shear stress system offers a robust platform for exploring the interplay between fluid dynamics and cellular mechanics. Our findings enhance the understanding of fluid shear stress effects on cell mechanics and have implications for biomaterials development and cell therapy strategies in cancer metastasis research.

## Author contributions

**Conceptualization:** Xinyao Zhao.

**Data curation:** Xinyao Zhao, Xiaolong Zhang.

**Formal analysis:** Xinyao Zhao.

**Investigation:** Fei Lei, Yaoxian Wang.

**Methodology:** Xinyao Zhao, Xiaolong Zhang, Yaoxian Wang.

**Project administration:** Xiaolong Zhang.

**Resources:** Xiaolong Zhang.

**Software:** Weikang Guo.

**Supervision:** Hui Yu, Yaoxian Wang.

**Validation:** Weikang Guo.

**Visualization:** Fei Lei.

**Writing – original draft:** Xinyao Zhao.

**Writing – review & editing:** Xinyao Zhao, Yaoxian Wang.

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
