## [Decision Letter · Decision Letter 0]

21 Jan 2025

PONE-D-24-57138Effects of Fluid Shear Stress Duration on the Mechanical Properties of HeLa Cells using Atomic Force MicroscopyPLOS ONE

Dear Dr. Wang,

Thank you for submitting your manuscript to PLOS ONE. After careful consideration, we feel that it has merit but does not fully meet PLOS ONE’s publication criteria as it currently stands. Therefore, we invite you to submit a revised version of the manuscript that addresses the points raised during the review process.

We look forward to receiving your revised manuscript.

Kind regards,

Kulwinder Kaur, Ph.D.

Academic Editor

PLOS ONE

Journal Requirements:

Please confirm at this time whether or not your submission contains all raw data required to replicate the results of your study. Authors must share the “minimal data set” for their submission. PLOS defines the minimal data set to consist of the data required to replicate all study findings reported in the article, as well as related metadata and methods (https://journals.plos.org/plosone/s/data-availability#loc-minimal-data-set-definition ).

If your submission does not contain these data, please either upload them as Supporting Information files or deposit them to a stable, public repository and provide us with the relevant URLs, DOIs, or accession numbers. For a list of recommended repositories, please see https://journals.plos.org/plosone/s/recommended-repositories .

Additional Editor Comments:

Reviewer 1: This manuscript investigates the impact of fluid shear stress (FSS) duration on the mechanical properties of HeLa cells, presenting valuable insights into the field. However, several aspects require further clarification and improvement to enhance the manuscript's scientific rigor and comprehensiveness.

1. For introduction, the authors should systematically summarize the key findings from previous research on the effects of FSS on cells, especially those relevant to cancer cells. Additionally, how do the current study's objectives build upon or address the limitations of these prior works? A more coherent connection would strengthen the introduction.

2. Were there any quality control measures in place to ensure the consistency and stability of the HeLa cell line throughout the experiments? The authors should explain why DMEM-H medium was specifically chosen over other commonly used media for culturing HeLa cells?

3. Although the dimensions and assembly of the parallel - plate flow chamber are described, the selection of these specific dimensions lacks justification. How were these dimensions determined? Were there any preliminary simulations or experiments to optimize the chamber design for the intended fluid shear stress applications? Additionally, it would be helpful to know more about the material of the flow chamber and its potential impact on cell adhesion and the fluid flow characteristics.

4. the authors could provide more details on the validation of the fluid shear stress calculation method. Were there any experimental measurements to corroborate the calculated values?

5. Were multiple cells measured for each experimental condition, and if so, what was the level of variation in the measured mechanical properties? Additionally, the choice of the Sneddon model for analyzing the force - displacement curves should be justified. Are there any limitations of this model when applied to HeLa cells, and were other models considered?

6. The manuscript describes the changes in cell morphology and mechanical properties separately. It would be interesting to explore if there is a direct correlation between the two. could the authors perform a correlation analysis between the cell height and the Young's modulus to determine if changes in one property are associated with changes in the other?

7. The authors discuss potential mechanisms underlying the observed changes in cell mechanics, such as cytoskeletal reorganization, alterations in cell membrane fluidity, and changes in intracellular signal transduction. However, these discussions are rather speculative. Are there any experimental data or references to support these proposed mechanisms?

8. the connection between the mechanical properties measured in vitro and the actual metastatic behavior of HeLa cells in vivo remains unclear. Could the authors discuss how these in vitro findings might translate to the in vivo context? Are there any in vivo studies that support the relationship between mechanical properties and metastasis?

9. Recent work in the field of Mechanical Properties of Cells should be cited in introduction, such as: https://doi.org/10.1016/j.biotechadv.2020.107648; DOI: 10.1017/S1431927622000290; DOI: 10.1039/D0LC00477D; doi: 10.3791/50497

Reviewer 2: The mansucript presents an experimental study on how cells change their mechanics after being exposed to hydrodynamic flow. This is an interesting and relevant subject, and the results are novel. That said, the study does suffer from some major issues that need to be improved, before publication can be recommendend. These are, in order of relevance:

1) It is not clearly said, but the way I understand the manuscript, each experiment is carried out for a single cell only. The error bars appear to stem from different measurement points on the same cell. Measuring AFM at different points on the same cell is highly important and a good point of the paper.

However, we can expect significant cell-to-cell variation as well. Therefore, it is highly important to make statistics on, at least, 5-10 different cells for each experimental situation.

2) I am missing a quantitative connection to earlier works. The HeLa cells used in the manuscript are a widely-used cell line and I am sure there exist earlier AFM (and possibly other) measurements of their elastic properties in the absence of flow. The authors should look for such data and compare their control experiment without flow to these works.

3) The shear stress calculation in 2.3 1) seems to be an approximation. The exact flow field for a rectangular geometry is well-known, but is a somewhat lengthy expression containing an infinite convergent sum as given in standard texts on hydrodynamics. Please state which approximations were made to arrive at the simple form given in 2.3 1).

Reviewers' comments:

Reviewer's Responses to Questions

**Comments to the Author**

1. Is the manuscript technically sound, and do the data support the conclusions?

Reviewer #1: Partly

Reviewer #2: Yes

2. Has the statistical analysis been performed appropriately and rigorously? 

Reviewer #1: No

Reviewer #2: Yes

3. Have the authors made all data underlying the findings in their manuscript fully available?

Reviewer #1: No

Reviewer #2: Yes

4. Is the manuscript presented in an intelligible fashion and written in standard English?

Reviewer #1: Yes

Reviewer #2: Yes

5. Review Comments to the Author

Reviewer #1: The mansucript presents an experimental study on how cells change their mechanics after being exposed to hydrodynamic flow. This is an interesting and relevant subject, and the results are novel. That said, the study does suffer from some major issues that need to be improved, before publication can be recommendend. These are, in order of relevance:

1) It is not clearly said, but the way I understand the manuscript, each experiment is carried out for a single cell only. The error bars appear to stem from different measurement points on the same cell. Measuring AFM at different points on the same cell is highly important and a good point of the paper.

However, we can expect significant cell-to-cell variation as well. Therefore, it is highly important to make statistics on, at least, 5-10 different cells for each experimental situation.

2) I am missing a quantitative connection to earlier works. The HeLa cells used in the manuscript are a widely-used cell line and I am sure there exist earlier AFM (and possibly other) measurements of their elastic properties in the absence of flow. The authors should look for such data and compare their control experiment without flow to these works.

3) The shear stress calculation in 2.3 1) seems to be an approximation. The exact flow field for a rectangular geometry is well-known, but is a somewhat lengthy expression containing an infinite convergent sum as given in standard texts on hydrodynamics. Please state which approximations were made to arrive at the simple form given in 2.3 1).

Reviewer #2: This manuscript investigates the impact of fluid shear stress (FSS) duration on the mechanical properties of HeLa cells, presenting valuable insights into the field. However, several aspects require further clarification and improvement to enhance the manuscript's scientific rigor and comprehensiveness.

1. For introduction, the authors should systematically summarize the key findings from previous research on the effects of FSS on cells, especially those relevant to cancer cells. Additionally, how do the current study's objectives build upon or address the limitations of these prior works? A more coherent connection would strengthen the introduction.

2. Were there any quality control measures in place to ensure the consistency and stability of the HeLa cell line throughout the experiments? The authors should explain why DMEM-H medium was specifically chosen over other commonly used media for culturing HeLa cells?

3. Although the dimensions and assembly of the parallel - plate flow chamber are described, the selection of these specific dimensions lacks justification. How were these dimensions determined? Were there any preliminary simulations or experiments to optimize the chamber design for the intended fluid shear stress applications? Additionally, it would be helpful to know more about the material of the flow chamber and its potential impact on cell adhesion and the fluid flow characteristics.

4. the authors could provide more details on the validation of the fluid shear stress calculation method. Were there any experimental measurements to corroborate the calculated values?

5. Were multiple cells measured for each experimental condition, and if so, what was the level of variation in the measured mechanical properties? Additionally, the choice of the Sneddon model for analyzing the force - displacement curves should be justified. Are there any limitations of this model when applied to HeLa cells, and were other models considered?

6. The manuscript describes the changes in cell morphology and mechanical properties separately. It would be interesting to explore if there is a direct correlation between the two. could the authors perform a correlation analysis between the cell height and the Young's modulus to determine if changes in one property are associated with changes in the other?

7. The authors discuss potential mechanisms underlying the observed changes in cell mechanics, such as cytoskeletal reorganization, alterations in cell membrane fluidity, and changes in intracellular signal transduction. However, these discussions are rather speculative. Are there any experimental data or references to support these proposed mechanisms?

8. the connection between the mechanical properties measured in vitro and the actual metastatic behavior of HeLa cells in vivo remains unclear. Could the authors discuss how these in vitro findings might translate to the in vivo context? Are there any in vivo studies that support the relationship between mechanical properties and metastasis?

9. Recent work in the field of Mechanical Properties of Cells should be cited in introduction, such as: https://doi.org/10.1016/j.biotechadv.2020.107648; DOI: 10.1017/S1431927622000290; DOI: 10.1039/D0LC00477D; doi: 10.3791/50497

6. PLOS authors have the option to publish the peer review history of their article (what does this mean? ). If published, this will include your full peer review and any attached files.

**Do you want your identity to be public for this peer review?** For information about this choice, including consent withdrawal, please see our Privacy Policy .

Reviewer #1: No

Reviewer #2: No

---

## [Author Response · Author response to Decision Letter 1]

26 Feb 2025

Dear editor and reviewers,

Thank you for offering us an opportunity to improve the quality of our submitted manuscript (Effects of Fluid Shear Stress Duration on the Mechanical Properties of HeLa Cells using Atomic Force Microscopy). We appreciated very much the reviewers’ constructive and insightful comments. In this revision, we have addressed all of these comments/suggestions. We hope the revised manuscript has now met the publication standard of your journal.

We highlighted all the revisions in red colour.

On the next pages, our point-to-point responses to the queries raised by the reviewers are listed.

Reviewer 1: This manuscript investigates the impact of fluid shear stress (FSS) duration on the mechanical properties of HeLa cells, presenting valuable insights into the field. However, several aspects require further clarification and improvement to enhance the manuscript's scientific rigor and comprehensiveness.

Comment 1. For introduction, the authors should systematically summarize the key findings from previous research on the effects of FSS on cells, especially those relevant to cancer cells. Additionally, how do the current study's objectives build upon or address the limitations of these prior works? A more coherent connection would strengthen the introduction.

Response: Thanks for the suggestions. We have revised this part in the manuscript to enhance its clarity and depth. Below is our response to the issues you raised and the corresponding modifications. Page 3, line 58 to 110. In the revised manuscript, we have expanded the introduction section to systematically summarize previous studies on the effects of FSS on cancer cells and clarified how the current research advances further based on these works. Through these revisions, we have enhanced the coherence and scientific rigor of the introduction, highlighting the novelty and significance of the current study in building upon prior research.

Comment 2. Were there any quality control measures in place to ensure the consistency and stability of the HeLa cell line throughout the experiments? The authors should explain why DMEM-H medium was specifically chosen over other commonly used media for culturing HeLa cells?

Response: Thank you for your question regarding the quality control measures for the HeLa cell line. The relevant modifications have been made in the manuscript on page 19, lines 397-402.

To ensure the consistency and stability of HeLa cells, all cells were obtained from the Cell Bank of the Chinese Academy of Sciences, ensuring a uniform genetic background and absence of contamination. To eliminate the impact of cell passage batch variations on the measurement results, all experiments were conducted using cells from the same passage. The cells were cultured under identical conditions, and the application of fluid shear stress in the in vitro system was performed in the same environment. Additionally, five independent experiments were carried out to ensure the reliability of the results. We strictly controlled the culture environment, monitored cell morphology, and excluded abnormal differentiation or apoptosis. All procedures, including cell culture and shear stress experiments, were carried out in a 37°C, 5% CO₂ incubator, with standardized passage density, trypsin digestion time, and medium replacement frequency.

In the revised manuscript, we have explained why DMEM-H medium was specifically chosen over other commonly used media for culturing HeLa cells (page 7, lines 137 to 140). DMEM-H was chosen as the culture medium because it is the standard for HeLa cell culture. Its high glucose concentration (4.5 g/L) meets the high metabolic demands of HeLa cells, better supports their growth and proliferation, prevents stress responses caused by nutrient deficiency, and maintains energy metabolism balance under shear stress conditions.

Comment 3. Although the dimensions and assembly of the parallel - plate flow chamber are described, the selection of these specific dimensions lacks justification. How were these dimensions determined? Were there any preliminary simulations or experiments to optimize the chamber design for the intended fluid shear stress applications? Additionally, it would be helpful to know more about the material of the flow chamber and its potential impact on cell adhesion and the fluid flow characteristics.

Response: Thank you for your attention to the design details of the parallel-plate flow chamber. Here is our detailed response to the questions you raised:

The parallel-plate flow chamber is a commonly used model for studying cellular mechanical properties in vitro and is widely employed to simulate the physiological environment of cells under dynamic fluid shear stress (FSS). In this study, we referred to the mature product design from GlycoTech, as shown in Figure 1, and selected a rectangular flow chamber with a width of 1 cm and a length of 4 cm. The main reason for choosing this size is that the larger experimental area (1 cm × 4 cm) can provide a greater number of cell samples, which facilitates the extraction of sufficient cells for subsequent experiments (such as gene expression and protein analysis) after shear stress application. This avoids the risk of experimental interruptions due to insufficient sample quantities in circular flow chambers.

Fig. 1: Parallel-Plate Flow Chamber from GlycoTech

To ensure the rationality of the flow chamber design and the accuracy of the experiments, we conducted computational fluid dynamics (CFD) simulations. The simulation results show that within the target shear stress range (e.g., 0.1-10 dyn/cm²), the flow velocity distribution in the flow chamber is uniform, with fully developed boundary layers and shear stress errors of less than 1%, meeting the experimental accuracy requirements. Combining the CFD results with the classical parallel-plate laminar flow theoretical formula further confirms the rationality of the dimension parameters.

The main body of the flow chamber is made of polycarbonate, which has the advantages of good biocompatibility, high optical transparency (convenient for microscopic observation), and strong chemical stability (tolerant to cell culture media and sterilization processes).

Although the material of the flow chamber may have potential effects on cell adhesion and fluid flow properties, all experimental groups in this study used flow chambers made of the same material, eliminating the interference of material differences on cell adhesion and fluid properties. This ensures a clear causal relationship between shear stress exposure time (the only variable) and cellular response.

By referring to mature designs, validating with CFD simulations, and strictly controlling material consistency, we ensured the functionality and experimental reliability of the parallel-plate flow chamber. The material properties of the parallel-plate flow chamber may affect initial cell adhesion and flow field boundary conditions, which provides an important direction for our future research. We will further explore the mechanisms by which material properties affect the cellular mechanical microenvironment.

Comment 4. the authors could provide more details on the validation of the fluid shear stress calculation method. Were there any experimental measurements to corroborate the calculated values?

Response: We thank you for your valuable feedback regarding the validation of the fluid shear stress calculation method. We fully agree that providing detailed validation is crucial to ensuring the reliability of the research results. Below is our response to the issues you raised and the corresponding modifications. Page 10, line 196 to 201. Page 16, line 338to 349.

To validate the fluid shear stress calculation method, we combined theoretical formulas with numerical calculations. We used the classical fluid shear stress theoretical formula for computation and compared the results with numerical calculations. The formula is as follows:

1) Fluid Shear Stress Calculation Formula:

: Shear stress, in dynes per square centimeter (dyn/cm2);

Fluid viscosity, in poise (P);

a: Channel height, in centimeters (cm);

b: Channel width, in centimeters (cm);

Q: Liquid flow rate, in milliliters per second (ml/s).

In this study, the channel height a=0.3 cm,a, the channel width  b=1cm, and the fluid viscosity  μ=1P. Based on the theoretical formula, we calculated the flow rates Q corresponding to four different shear stress levels: 5, 10, 15, and 20 dyn/cm². The corresponding flow rates were 0.075, 0.15, 0.225, and 0.3 ml/s, respectively.

For each flow rate parameter, we computed the theoretical shear stress values and compared them with the numerical calculation results. The specific results are shown in the Table 1 below:

Table 1. Comparison of Theoretical and Numerical Fluid Shear Stress Results

Flow Rate Q (ml/s) Theoretical Shear Stress

τw (dyn/cm²) Numerical Shear Stress

τw (dyn/cm²) Error

0.075 5 5.01 �1%

0.15 10 10.04 �1%

0.225 15 15.05 �1%

0.3 20 20.05 �1%

As shown in Table 1, the error between the theoretical and numerical results is less than 1% in all cases. This demonstrates that our fluid shear stress calculation method is highly accurate and can be reliably applied in fluid shear stress experiments.

Comment 5. Were multiple cells measured for each experimental condition, and if so, what was the level of variation in the measured mechanical properties? Additionally, the choice of the Sneddon model for analyzing the force - displacement curves should be justified. Are there any limitations of this model when applied to HeLa cells, and were other models considered?

Response: We thank the reviewer for their question. During the experiments, we actually measured multiple cells under each experimental condition. However, this was not explicitly stated in the experimental methods section of the original manuscript. To obtain statistically significant results, we performed morphological imaging on one group of cells and measured force curves for five different groups of cells under each experimental condition (the experimental methods section has been updated in the manuscript. Page 19, line 397 to 402�Page19�line406 to 411). The Young's modulus measurement results for multiple groups of cells, obtained using AFM, are shown in Fig. 2

Fig. 2 Measurement of Young's modulus for multiple groups of cells using AFM

We measured force curves for five groups of cells under each experimental condition and fitted 20 force curves from each group to determine the Young’s modulus of the cells. �Page22,line457-459.�The experimental results indicate that under identical fluid shear stress conditions, the Young’s modulus values of the cells tend to decrease with increased duration of exposure. As shown in Fig.3.

Fig. 3 Measuring the Young's modulus of Hela cells subjected to different durations of fluid shear stress by Atomic Force Microscopy (AFM)

In the revised manuscript, we have explained the rationale for choosing the Sneddon model to analyze the force-displacement curves, and discussed the potential limitations of this model when applied to HeLa cells, as well as whether other models were considered. The detailed explanation is located on Page 13, lines 263-278.

The force curve is converted into an indentation curve based on the contact point on the force curve. Subsequently, the indentation curve is fitted using theoretical models to obtain the Young's modulus of the cells. Currently, the main models available for extracting the Young's modulus of cells are the Hertz-Sneddon, JKR (Johnson-Kendall-Roberts), and DMT (Derjaguin-Muller-Toporov) models. The Hertz-Sneddon model neglects the forces in the contact area between the tip and the cell, such as electrostatic forces, adhesive forces, and frictional forces. In contrast, the JKR and DMT models consider the adhesive forces within the contact area. The JKR model is suitable for situations where the tip is large and the adhesive force between the tip and the sample is significant, while the DMT model is applicable when the tip is small and the adhesive force is minimal. However, the most widely used model in practice is the Hertz-Sneddon model. The Hertz-Sneddon model is based on a series of assumptions about the sample being probed, such as isotropy, homogeneity, linear elasticity, axisymmetry, infinite thickness, and smooth surface. Although cells do not strictly meet these conditions, studies have shown that the Hertz model is applicable when the indentation depth is less than 10% of the sample thickness. Therefore, this study employs the Sneddon model to fit the indentation curves, thereby obtaining the cellular Young's modulus.

Comment 6. The manuscript describes the changes in cell morphology and mechanical properties separately. It would be interesting to explore if there is a direct correlation between the two. could the authors perform a correlation analysis between the cell height and the Young's modulus to determine if changes in one property are associated with changes in the other?

Response: We thank the reviewer for their attention to the correlation between cell morphology and mechanical properties. We agree on the importance of exploring the relationship between cell height and Young's modulus and have included preliminary analysis and discussion in the revised manuscript. Page 24, line 504 to 513. Below is our response to the issues you raised and the corresponding modifications.

This study found that both cell morphology and mechanical properties undergo significant changes under fluid shear stress (FSS). The experimental results indicate that as the duration of FSS increases, HeLa cell height decreases, the morphology becomes more flattened, and the Young's modulus significantly decreases. This phenomenon is consistent with literature reports. For example, Novak et al. (as cited in reference [42] of our manuscript) observed that breast cancer cells exposed to shear stress in the tumor microenvironment (TME) exhibited significantly larger cell area, lower roundness, and a more motile phenotype. These stimulated cells were more proliferative than static controls and showed higher resistance to the anti-cancer drug paclitaxel. These results suggest that pulsatile shear stress promotes proliferation, invasive potential, chemotherapy resistance, and PLAU signaling in breast cancer cells. Fan et al. (as cited in reference [43] of our manuscript) observed through micropatterned cell culture that cells subjected to mechanical stimulation exhibited reduced height, a more flattened morphology, and cytoskeletal reorganization leading to a decrease in Young's modulus, making the cells softer. These changes in mechanical properties further influence cell functions such as migration, invasion, and nuclear gene expression.

To more deeply quantify the relationship between cell morphology and mechanical properties, we plan to systematically analyze multiple groups of cells under the same experimental conditions in future studies. This will provide more direct experimental evidence to reveal the interaction between cell morphology and mechanical properties.

Comment 7. The authors discuss potential mechanisms underlying the observed changes in cell mechanics, such as cytoskeletal reorganization, alterations in cell membrane fluidity, and changes in intracellular signal transduction. However, these discussions are rather speculative. Are there any experimental data or references to support these proposed mechanisms?

Response: We thank you for your valuable comments on the potential mechanisms underlying the changes in cellular mechanical properties. In the revised manuscript, we have added relevant references and discussions to enhance the credibility of these mechanisms. Page 3, line 58 to 71. Page 23, line 482 to 503, page 24, line514 to 518.

The following references and research findings support the potential mechanisms of changes in cellular mechanical properties:

1.Luo et al. (as cited in reference [40] of our manuscript): The study demonstrated that the stiffness of cancer cells and its changes during metastasis are crucial for understanding the pathophysiology of cancer cells and the mechanisms of c

---

## [Editor Report · Decision Letter 1]

4 Mar 2025

Effects of Fluid Shear Stress Duration on the Mechanical Properties of HeLa Cells using Atomic Force Microscopy

PONE-D-24-57138R1

Dear Dr. Wang,

We’re pleased to inform you that your manuscript has been judged scientifically suitable for publication and will be formally accepted for publication once it meets all outstanding technical requirements.

Kind regards,

Kulwinder Kaur, Ph.D.

Academic Editor

PLOS ONE
---

## [Editor Report · Acceptance letter]

PONE-D-24-57138R1

PLOS ONE

Dear Dr. Wang,

I'm pleased to inform you that your manuscript has been deemed suitable for publication in PLOS ONE. Congratulations! Your manuscript is now being handed over to our production team.

Kind regards,

on behalf of

Dr. Kulwinder Kaur

Academic Editor

PLOS ONE